# Adsorption Mechanism of Patulin from Apple Juice by Inactivated Lactic Acid Bacteria Isolated from Kefir Grains

**DOI:** 10.3390/toxins13070434

**Published:** 2021-06-22

**Authors:** Pascaline Bahati, Xuejun Zeng, Ferdinand Uzizerimana, Ariunsaikhan Tsoggerel, Muhammad Awais, Guo Qi, Rui Cai, Tianli Yue, Yahong Yuan

**Affiliations:** 1College of Food Science and Engineering, Northwest A&F University, Yangling 712100, China; bahatipa@nwafu.ed.cn (P.B.); zxj1990@nwafu.edu.cn (X.Z.); ariunsaikhan@nwafu.edu.cn (A.T.); guoqiqi@nwafu.edu.cn (G.Q.); cairui@nwsuaf.edu.cn (R.C.); yuetl305@nwafu.edu.cn (T.Y.); 2State Key Laboratory of Crop Stress Biology in Arid Areas, College of Agronomy, Northwest A&F University, Yangling 712100, China; uziferd@nwafu.edu.cn; 3College of Plant Protection, Northwest A&F University, Yangling 712100, China; awaismuhammad@nwafu.edu.cn

**Keywords:** mycotoxin, patulin, lactic acid bacteria, kefir, apple juice, binding mechanism

## Abstract

In the food industry, microbiological safety is a major concern. Mycotoxin patulin represents a potential health hazard, as it is heat-resistant and may develop at any stage during the food chain, especially in apple-based products, leading to severe effects on human health, poor quality products, and profit reductions. The target of the study was to identify and characterize an excellent adsorbent to remove patulin from apple juice efficiently and to assess its adsorption mechanism. To prevent juice fermentation and/or contamination, autoclaving was involved to inactivate bacteria before the adsorption process. The HPLC (high-performance liquid chromatography) outcome proved that all isolated strains from kefir grains could reduce patulin from apple juice. A high removal of 93% was found for juice having a 4.6 pH, 15° Brix, and patulin concentration of 100 μg/L by *Lactobacillus kefiranofacien*, named JKSP109, which was morphologically the smoothest and biggest of all isolates in terms of cell wall volume and surface area characterized by SEM (Scanning electron microscopy) and TEM (transmission electron microscopy). C=O, OH, C–H, and N–O were the main functional groups engaged in patulin adsorption indicated by FTIR (Fourier transform–infrared). E-nose (electronic nose) was performed to evaluate the aroma quality of the juices. PCA (Principal component analysis) results showed that no significant changes occurred between control and treated juice.

## 1. Introduction

Patulin is a hazardous mycotoxin to human health. It is known to originate from more than 60 species of fungi belonging to more than 30 genera, principally by *penicillium expansum,* which commonly occurs in rotting apples. Patulin is toxic to human beings, animals, and crops exposed to it, and it affects economic growth [1,2,3]. Historically, patulin was screened as an antibiotic during scientific endeavors in the 1940s [4]. Patulin was shown to have the ability to inhibit several harmful bacteria, and has been used for therapeutic purposes [5]. However, many researchers declared its adverse health effect [6]. Most research on patulin confirmed genotoxicity [2,7]. The report made by Puel et al. [8] showed that the source of mutagenic effects caused by patulin is connected to its high reactivity to the thiol groups of proteins and glutathione (GSH). These mutagenic effects are mostly in low glutathione levels of the cells, causing the damage of the chromosome and generating a micronucleus. The toxicity mechanisms of PAT are due to the covalent bonds of electrophilic chemicals made by the reaction of amino acids, including histidine, lysine, and cysteine with PAT [9]. [10] reported that the activation of the Rpn4 transcription factor leads to the overexpression of the Rpn4 gene and indicates the genotoxic effect. The expression of different autophagy markers (LC3-II and LC31) are increased by PAT, leading to activation of the autophagic system, which later causes the degradation of the cytoplasmic protein and causes DNA damage [11]. PAT noticeably affects the quality of apples, and is responsible for the soft rot of several fruits [1]. PAT is mostly distributed in nature by blue mold disease originating from *P. expansum* found in infected fruits, especially apples [12,13]. Patulin is slightly stable at pH < 5.5 and soluble in water [14]. Being a polyketide lactone in nature, PAT generates heat resistance properties, resulting in its stability during and after heat treatment, such as pasteurization [8,15,16,17,18,19,20,21]. Apart from citrus fruits and grapes, apples represent the main produced fruit in the USA. After grapes, apples are ranked the second most produced fruit in China [10]. These were confirmed by [9], who reported apples to be among the important fruits processed worldwide. Yue et al. [13] and Moake et al. [18] criticized many different fruit plants for using poor quality raw materials, such as rotten, damaged, and bruising apples, which promote PAT-producing fungal growth and PAT accumulation. PAT has been found in various food commodities, but mostly in apples and apple products [6], and the indications show that infants’ and young children’s health is particularly exposed to the threat posed by PAT contamination of apples and apple-based products [13]. Preventative measures and maximum levels (MLs) for PAT contamination have been established: the Food and Drug Administration (FDA) in the US fixed a maximum level of 50 μg/L, and the FAO/WHO has set 0.4 μg/kg body weight/day as the allowable maximum intake. Certain countries agreed on a maximum limit of 25 to 35 µg/L [2,22]. The European Union set a limitation of PAT concentration to 50 μg/kg in spirit drinks, fruit juice, and cider derived from apples, and 10 micrograms/kg and 25 μg/kg limits for young children and in solid apple products, respectively [16]. Despite the abovementioned standards, inspections have continued to find that PAT in some commercial apple-based products from different countries, including developed countries, such as the USA, Australia, and China, exceeds the maximum limits [23,24].

To prevent, reduce, and/or detoxify contaminated apple juice, several physical and chemical strategies have been tested [13,25,26,27]. Physical methods, including refrigeration, high-pressure water washing, manual picking, juice clarification, filtration and/or adsorption, radiation, and pasteurization, were widely used to prevent and inhibit PAT-producing fungi growth in apples and their derivatives [18]. Although these cited methods work well, their effectiveness is limited due to high costs and great manpower [14,18]. Heat treatment has been tested to stop *P. expansum* actions by exposing it to 45 °C water for 10 min [20]. The humidity of 0.99 and above was declared to be a factor favoring the growth of *P. expansum*. According to [21], the decrease of humidity from 0.99 to 0.85 can reduce the production of PAT. However, the abovementioned methods are inappropriate for use at large scales.

Many chemical substances, including calcium-d-pantothenate, ascorbic acid, ammonia, calcium-d-pantothenate, pyridoxine and thiamine hydrochloride, sulfur dioxide, sulfur dioxide, hydrochloride, and ozone, have been tested to remove PAT from different food products, notably apple juice [16,28]. The ability of potassium permanganate and ammonia to remove 99% of PAT was confirmed and believed to be more effective [14,29]. These chemical strategies showed reliable results in PAT contamination control from different food products, including apple juice. However, the basic sensory and quality attributes have not been maintained for some approaches [30]. In addition, they have remarkable disadvantages for food safety and product quality, such as chemical hazard introduction and nutritional losses [11,31,32]. Moreover, the unspecified byproduct generation declared their imperfection [31,33]. However, basic sensory and quality attributes have not been maintained for some approaches; in addition, the abovementioned strategies have remarkable disadvantages on food safety, such as chemical hazards introduction, and nutritional losses lead to poor quality products. Moreover, the effectiveness is limited due to high costs [6,24,25,26]. Current research has reported the production of mycotoxin patulin in apples infected with *Paecilomyces niveus*.

Up to the present, the amount of research on PAT removal has been expanded worldwide, and most experiments have been concerned with the use of microorganisms in PAT adsorption in food safety and quality control. The biological techniques in PAT control refer to the use of enzymes and/or inactive or active harmless microorganism (fungi, yeast, and bacteria) to prevent, inhibit, reduce, or remove PAT produced by fungi in food products, including apple and apple-based products, through adsorption or degradation [32]. The use of several antagonistic microorganisms, including active and inactivated cells to degrade or adsorb PAT, were highly studied and confirmed by different researchers to be the effective way to remove PAT [1,14,34].

The potential of lactic acid bacteria (LAB) in removing PAT from diverse food, notably apple and its derivatives, and the mechanism involved have been sufficiently investigated by many authors and believed to be competent, safe, and reliable for PAT adsorption and/or degradation from apple juice, with no detrimental effect on juice quality parameters [30,31,34,35,36,37,38,39,40]. The appreciable benefits of *L. casei*, such as bad bacteria, mold growth, and tumor inhibition, in addition to lipid metabolism, antioxidant capacity, and immunity improvement, are well-known. Moreover, this bacteria species showed the ability to decrease weight, prevent type 2 diabetes, and protect the liver [14] Recently, the ability and mechanism of *L. casei* YZU01 to remove PAT from apple juice was investigated after contaminating juice with PAT concentrated to 10.9 μg/mL [14]. The outcome indicated that the PAT was completely degraded after 36 h in raw apple juice. The authors concluded that, within 48 h, *L. casei* YZU01 had a potential ability to remove 95% of PAT contamination in apple juice. Microorganisms such as fungi, yeast, and bacteria have been confirmed to remove and/or reduce patulin from different juices and aqueous solutions through an adsorption mechanism [24,34]. Several microorganisms, including lactic acid bacteria, were investigated to be efficient, safe, and reliable for PAT adsorption from apple juice [24,37,41].

Kefir grains are composed of a complex of microbial species classified into the following groups: homo and heterofermentative lactic acid bacteria, lactose, and non-lactose assimilating yeast [42]. Kefir grains are predominated by both lactic and acetic acid bacteria, yeast, and fungi. Tibetan kefir is a kind of milk kefir mostly found in China, which is composed of *lactobacillus, lactococcus*, and yeast. Lactic acid bacteria (LAB) are the microbial consortium found in kefir [43]. Moreover, the microorganism consortium of kefir grains showed the ability to absorb 82 to 100% of mycotoxin [44,45]. Initial PAT concentration, contact time, pH, and temperature addition to the adsorption mechanism of PAT were studied [17,24,46,47]. However, there are not enough citations showing the adsorption mechanism samples having different °Brix were provided, and the adsorption mechanism of PAT needs to be clearer [35].

This research mainly aimed to identify and characterize the kefir grain cells with a high ability to reduce or remove PAT from apple juice with different °Brix and pH, to assess its binding mechanism during adsorption, and to provide a comprehensive method for apple juice detoxification without any detriment to quality and safety of the end product. DNA sequencing was used to identify the main kefir grain cells (LAB) involved in the adsorption with a high potential ability to adsorb PAT in apple juice. High-performance liquid chromatography (HPLC) was performed in the identification and quantification of the adsorbed PAT. Scanning electron microscopy (SEM) and transmission electron microscopy (TEM) were used to characterize bacterial cells. Fourier transform–infrared (FT-IR) analysis was applied to point out the important functional groups and the possible binding sites of the tested bacterial strains. The quality attributes of apple juice were tested; E-nose (an electronic nose) was applied for volatile compounds analysis.

## 2. Results

### 2.1. Strains Identification

All five isolated strains from kefir grains were identified using molecular methods according to 16S rRNA gene sequencing, and then the nucleotide sequences of bacteria were submitted to the NCBI database. The results indicated that lactic acid bacteria strains WKLP10 and WKLF71 isolated from water kefir grains (Tibicos) had 100% nucleotide homology with *Lactobacillus plantarum* and *Lactobacillus Lactobacillus Fermentum*, respectively, except for WKLF76, identified as *Lactobacillus Fermentum*, which had 99% homology to the GenBank sequences. Both JKSP1 and JKSP109 were identified as *Lactobacillus kefiranofaciens* (99% homology to the GenBank sequences) and were isolated from milk kefir grains (Tibetan kefir grains). The 16S rRNA sequences of the five strains WKLP10, WKLF71, WKLF76, JKSP109, and JKSP1 were deposited in GenBank under the accession numbers MZ359811, MZ021346, MZ021345, MW828338, and MZ021343, respectively. In this study, *Lactobacillus Plantarum* WKLP10, *Lactobacillus Fermentum* WKLF71, and *Lactobacillus fermentum* WKLF76 isolated from Tibicos were named LP10, LF1, and LF6, respectively. *Lactobacillus kefiranofaciens* JKSP109 and *Lactobacillus kefiranofaciens* JKSP1 isolated from Tibetan kefir grains were named LK3 and LK2, respectively.

### 2.2. HPLC Analysis

The binding effect was strong for almost all of the isolates from both Tibetan kefir grains (milk kefir grains) and Tibico (water kefir grains) consortium in all juices, named AJE, AJD, and AJC (apple juice having 15 °Brix, 4.6 pH; 12.5 °Brix, 3.8 pH; and 10.3 °Brix, 3.8 pH), respectively, except LP10, which only adsorbed 12% (Figure 1A). High PAT removal of 93% was in AJE (juice having 4.6 pH, 15 °Brix) and a PAT concentration of 100 μg/L by *lactobacillus kefiranofacien* JKSP109, named LK3, followed by LK2, LF1, LF6, and lastly LP10. In the juice samples contaminated at 200 µg/L, *Lactobacillus kefiranofacien* JKSP1, named LK2, showed the ability to adsorb 55.67% of PAT from AJE, according to Figure 1B. Besides LP10, which showed low ability in PAT adsorption, Tibetan kefir grains and Tibicos LAB strains LK2, LK3, LF1, and LF6, respectively, had shown different abilities in PAT adsorption with no significant difference in ADJ, significant in AJC, and highly significant in AJE, shown by LF6 at *p* ≤ 0.05 (Figure 1A). PAT concentration was also one of the important factors influencing PAT removal.

The adsorption ability was significantly increased by lowering the PAT concentration from 200 µg to 100 µg/L (Figure 1A,B). This was remarkable for all used bacteria strains in all juices where a sharp increase of the adsorption amount by reducing PAT concentration occurred, except LP10, which showed no significant removal between the two concentrations. When the concentration of PAT decreased from 200 µg/L to 100 µg/L, the adsorption capacity of LK3 increased from 49.4 to 93.2%, 18.5 to 55.6%, and from 23.6 to 63.9% in AJE, AJD, and AJC, respectively.

### 2.3. The Effect of the Adsorption Process on °Brix and pH

After the absorption processes, the pH and °Brix of the treated juices were assessed. Figure 2 shows the effect of the adsorption process on the pH and °Brix of treated apple juices at 100 and 200 µg/L PAT. No significant differences were found in the values for pH and °Brix based on (*p* ≤ 0.05) between the treated juices (juices with bacteria and PAT after adsorption) and the control (juice without bacteria and PAT) in 100 µg/L and 200 µg/L PAT concentration.

### 2.4. Juice Aroma Analysis

The aroma of apple juice with different °Brix and pH treated by different bacteria strains was thoroughly determined by E-nose (electronic nose) (Figure 3a,b). The variance of PC1 and PC2 accounted for 42.2% and 33.3% of the total variance, respectively, and the sum of their variance was up to 76.3% of the PCA results of the juice aroma. E-nose was performed to determine the aromas of apple juice with different °Brix and pH treatments by different kefir grain cells (LK2, LK3, LF1, and LF6). AJC and AJD were located at the positive part of PC1, while AJE is located in the negative part of PC1. W3C and W5C react to alkanes, aromatic compounds, and less polar compounds. These two sensors were high in AJE, opposite of AJC, which retained W3S and W2W sensitive to alkane (methane), aromatic compounds, and sulfur organic compounds, respectively. W1W, W1C, W1S, W2S, W6S, and W5S were greatly developed in AJD. In contrast, AJE showed the lowest signal response of these two sensors. The intensity of the W3S sensor was significantly reduced in both AJC and AJD. Apple juice (AJE) had the lowest value for the W1W sensor’s intensity, but there was no difference between AJC and AJD.

### 2.5. Morphological Analysis of Bacterial Cells by SEM and TEM

Morphological differences (shape and surface) between LP10, LF6, LF1, LK2, and LK3 were observed using SEM (Figure 4A). No differences between shapes were observed. There was a remarkable rough skin on LF and LP10, but no obvious differences between PAT exposed and non-exposed bacterial cells were observed in either surface or shape morphologies. The smoother the bacterial cell wall exterior, the higher the adsorption ability. Cell sizes were well-distinguished by size, but with common similarities in the shapes. TEM performance is shown in Figure 4B,C.

The adsorption process has shown that the bigger the cell wall volume and the surface area are, the higher the adsorption ability, as shown in Table 1. Cell wall volume and surface area values were plotted (Figure 5).

### 2.6. FTIR Analysis

The potential functional groups and the possible adsorption sites related to the adsorption of patulin for all strains in different juices were identified by FTIR analysis. Table 2 shows the FTIR spectra of all used lactic acid bacteria strains in AJE (apple juice: 15 °Brix and pH = 4.6). Some peak intensities were significantly changed to high and short for all bacteria strains, and some band spectra were shifted (Appendix A). The band absorbance, which appears on the bacteria cell wall, was assigned and matched with the vibration modes of the chemical bonds. Before PAT was loaded, the bacteria cell wall showed some different peaks corresponding to different functional groups. Figure 6 shows an example of LK3 used in different juices before and after PAT was loaded; at 3369.65866 cm^−1^, the broad and strong peak corresponded to the symmetrical and asymmetrical stretching vibrations of the O-H bond, H-bonded alcohols, and phenols. The presence of the alkane group was confirmed by the appearance of medium peaks at 2960.7476 cm^−1^ and 2929.8864 cm^−1^ of the alkane C-H bond stretching. The amide I peak presented at 1656.8614 cm^−1^ was governed by the stretching vibrations of the C=O. The peak presented at 1537.27416 cm^−1^ corresponded to N-H amide 2. The medium peak formed at 1450.477 cm^−1^ confirmed that the presence of alkane was attributed to CH2 bending vibration and the peak at 1394.5410 cm^−1^ of C-H bending. The C-N stretching vibration of aliphatic amines was observed as a medium at 1236.3773 cm^−1^. The strong peak at 1064.71184 cm^−1^ was considered as the coupling of C-O bond stretching devoted to alcohols, carboxylic acids, esters, and ethers. The medium peak at 570.93243 cm^−1^ had a C-Br stretching bond, indicating the presence of alkyl halide.

## 3. Discussion

In our study, the result showed that *Lactobacillus kefiranofaciens* was the only dominant bacterial species in Tibetan kefir grain. The results agreed with Wang et al. [48], who confirmed the domination of *Lactobacillus kefiranofaciens* in Tibetan kefir grains used in China after isolation based on culture-dependent and culture-independent methods. The same result occurred with Korsak et al. [49] and Lu et al. [50]. *Lactobacillus plantanum* and *Lactobacillus fermentum* were isolated from water kefir grains (Tibicos); the same results were found by Angelescu at al. [51]. However, these bacteria are mostly found in milk kefir grains and other dairy products [52].

The adsorbents showed a high capacity of PAT removal in AJE (juice with 15 °Brix, 4.6 pH) compared to AJC (juice with 10.3 °Brix, 3.8 pH) and AJD (juice with 3.8 pH and 12.5 °Brix). The highest adsorption of 93% was achieved in AJE at a 100 μg/L PAT concentration by *Lactobacillus kefiranofaciens* strain JKSP109, named LK3 (Figure 1), which has a higher surface area and cell wall volume (Table 1) and smooth surface (Figure 3a), followed by LK2, LF1, LF6, and LP10 with the adsorption capacities of 92%, 89%, 64%, and 10%, respectively. This could be more influenced by pH level than by °Brix. This agrees with [53]. In his research, he found that the amount of adsorbed PAT increased from 28.87 to 76.95% with an increase in pH from 3.0 to 4.5. A similar result was reported by Guo at al. [53] in the experiments conducted on milk. He observed that the increase of pH from 3 to 4.8 also raised the adsorption amount of AFB1, OTA, and ZEA by kefir microbial consortium from 6%, 5%, and 0% to 82%, 94%, and 100%, respectively. The maximum PAT removal was observed at pH = 3.0 to 4.0 [38]. On the other hand, the adsorption was higher in AJC than AJD having the same pH and different °Brix; this could have been influenced by the juice viscosity. The same result was found by N.L. Leggott et al. [54], who reported the difference of PAT removal in two different °Brix juices, where the achievement of adsorption level in 20 °Brix needed a double absorbent used in 12 °Brix. This could have been influenced by the increase in juice viscosity. The results also showed that the toxin reduction was strain- and PAT concentration-dependent [55,56]. The adsorption ability was significantly increased by lowering the PAT concentration from 200 µg/L to 100 µg/L (Figure 1A,B). The results agree with Hatab et al. [37], who reported that the increase of initial PAT concentration decreased the loading capacity.

Many researchers have already evaluated the effect of inactivated cell addition on the juice characteristics, such as °Brix, titratable acidity, total sugar, color value, and clarity [35,41]. The results suggested that its application in PAT juice detoxification was effective, with no harmful effects on the quality parameters of apple juice. In our study, after batch adsorption, the °Brix, pH, and aroma analysis of treated apple juice was assessed (Figure 2 and Figure 3). The result confirmed the fact that all used inactivated LAB cells in apple juice samples with different °Brix and pH did not affect the quality of the juice after statistical analysis (*p* < 0.05) of the aromatic compound.

PCA experiments have been performed for the recognition of several kinds of juice odors, and were especially directed at finding the difference between the juices, even if they belonged to the same brand [57,58]. E-nose technology has been widely used in the food field and has been studied in the field of food science, such as fruits and vegetables [59]; as shown in the graph (Figure 3a), the samples with different °Brix could be separated from each other. AJC and AJD were located on the positive part of PC1, indicating that there are some similarities between AJC and AJD. As indicated above, AJC and AJD have the same pH and different °Brix of 10.3 and 12.5; thus, their aromatic compound is relatively the same, unlike AJE, located in the negative part of PC1, indicating the significant differences between AJC and AJE. W3C is sensitive to aromatic compounds of ammonia, while W5C reacts to alkanes, aromatic compounds, and less polar compounds. These two sensors were high in AJE, unlike AJC, which retained a W3S sensor, sensitive to alkane (methane) and a W2W sensor, sensitive to aromatic compounds like sulfur organic compounds. On the other hand, sensor W1W, highly developed in AJD, reacted on many terpenes, organic sulfides, and inorganic sulfide (H2S), which were poor in ADJ. These were the same findings as Biasto et al. [60], who concluded that high °Brix content apple juice was reduced in terpenes. W1C, W1S, W2S, W6S, and W5S were greatly developed in AJD. PCA of E-nose data for apple juice with different °Brix and pH was remarkably different from each other. However, the juices treated with different bacterial strains were about the same as their control, implying that bacteria stains added to the juice for adsorption purposes did not affect the quality of the juice. Based on these findings, the addition of inactivated LAB cells in juice PAT detoxification might be a good choice, because inactivated cells do not apparently affect the aroma of the juice.

The TEM results showed that LK3, which had the highest surface area and cell wall volume (Figure 4B), had the highest capacity to adsorb PAT from the apple juice, followed by LK2, LF1, LF2, and, lastly, LP10, having the smallest surface area and volume and greater roughness (Table 1). This suited the trend that increasing adsorptive properties were usually associated with an increase in the surface area [55]. Besides this, our SEM results in (Figure 4A) showed remarkable differences in the cell wall appearance. LK3 showed a high adsorption ability and had the smoothest skin, followed LK2, LF1, LF2, and, lastly, LP10, which had the roughest skin. Many types of research are recommended to prove the present findings.

LAB was described as adhering to diverse mycotoxins, including PAT, and is significant [35,56,61,62]. The target of our research was to see if PAT could be extracted from apple juice using LAB. Our outcomes were positive (Figure 1). The same conclusions were addressed by [63,64]. Moreover, all of the used strains in this study, named LK2, LK3, LF1, and LF6, were successful in removing PAT by 93%, 92% 82%, and 67%, respectively, except LP10, which showed a low ability to adsorb patulin from apple juice. Our findings concur with many researchers’ results [45,64,65]. Our results showed that PAT adsorption was contingent upon the bacteria’s specifications, including the size of the bacteria, as has been illustrated [28,36,66]. Wang et al. [55] found that the efficiency of LAB strains to adsorb PAT was due to the amino acid and starch components of the cell wall.

Changes in absorption bands and peak intensity at 3369.65866, 1656.8614, and 1064.71184 cm^−1^ confirmed the participation of O-H carboxyl acid, alcohol phenol, C=O amide I, and a C-O stretching bond for polysaccharides, respectively, which indicated the participation of these groups in PAT adsorption. These changes occurred in all juice samples. There was a particular change at 2960.7476 cm^−1^ corresponding to C-H stretching; whereas a peak at 2960.7476 cm^−1^ completely disappeared for all juice samples. The peak 1450.477 cm^−1^ was attributed to CH2 bending, and vibration shifted to 1448.5482 cm^−1^ in AJE and AJD, except AJC showing that the CH2 groups were involved in the PAT adsorption. Increasing in peak intensity at 1064.71184 cm^−1^ and having strong asymmetrical C-O stretching (alcohols, carboxylic acids, esters, ethers) occur, followed by a slight change of wavenumbers at 1062.783 cm^−1^, confirmed the assumption that polysaccharides were also involved in the PAT adsorption in all juice samples [55].

## 4. Conclusions

The health risks posed by PAT necessitate its control and removal from apple products, creating a demand for food processing techniques capable of removing PAT. Pasteurization is a technique commonly used in food processing, including juice manufacturing, to ensure the destruction of enzymes and certain microorganisms, thus ensuring the safety of food and prolonging product shelf life. However, it cannot remove PAT, which is already present. The target of our research was to assess if PAT could be adsorbed from apple juice using different heat-inactivated LAB strains isolated from kefir grains in addition to the previous report. Our outcomes were positive. PCA results showed that the PAT adsorption was effective without affecting the quality of the juice. E-nose applications of these findings are important for quality control and quality assurance in terms of product quality and aroma assessment. The inactivated LAB cells, which had higher specific surface area, cell wall volume, and smoother surfaces, were more likely to have a higher capacity to adsorb PAT from apple juice and vice versa. Our FTIR results supported the conclusion that the polysaccharides and proteins of cell walls were important components in the adsorption of PAT. Our results concluded that using inactivated lactic acid bacteria isolated from kefir grains is very effective and safe for the juice consumers, without any health threat. Furthermore, the process is environmentally friendly. Thus, heat-inactivated *Lactobacillus kefiranofaciens* JKSP1, *Lactobacillus kefiranofaciens* JKSP109, *Lactobacillus fermentum* WKLF71, and *Lactobacillus Fermentum* WKLF76 are recommended to be used to remove PAT contamination in apple juice through adsorption.

## 5. Materials and Methods

### 5.1. Materials

Standard patulin was given by the university (NWAFU) Yangling, Shaanxi, China. Acetonitrile and ethyl acetate used for HPLC analysis were purchased from Sigma-Aldrich (St. Louis, MO, USA). MRS and other chemicals were bought from Chemical Reagents Company, Yangling, Shaanxi China.

### 5.2. Kefir Grains

Kefir grains were produced according to Taheur et al. [44] and Zanirati et al. [67]; milk Kefir grains (Tibetan kefir grais) were grown in ultra-high temperature cow milk and incubated at room temperature for three days. Afterward, the grains were washed with sterilized distilled water to remove the clotted milk. The grains were re-inoculated into sterile milk, followed by incubation at room temperature for two days. The process was repeated seven times for better growth. At the same time, kefir grains (Tibicos) were grown in a brown sugar solution according to the protocol, where 25 g of brown sugar was added to 200 mL of boiled water and cooled down to room temperature. Then, water kefir grains (10% *w/v*) were inoculated and kept at room temperature for two days. The grains were sieved and rinsed with sterile distilled water. This was repeated five times for better growth, followed by short-term storage at 4 °C.

### 5.3. Isolation and Enumeration of LABs from Grains

The method of Taheur et al. [44] was used with a slight modification; 10% *w/v* of both Tibetan and Tibicos kefir grains were ground separately using a mortar and pestle (instead of using a stomacher) and suspended in sterile saline (0.85%) solution in a different conical flask. The samples were performed in triplicates, serial decimal dilutions were prepared in the same diluent, and 0.1 mL was inoculated by surface spreading on specific solid media (MRS-Agar) and incubated at 37 °C for three days. After incubation, there was the enumeration of the result colonies, and the counts were demonstrated as the decimal logarithms of the colony-forming units per gram (log cfu/g). The isolated colonies were individually selected from the plates of each kefir grain cultivated on de Man Rogosa Sharpe agar (MRS) by streak plate purification in triplicate, and then re-incubated in the same condition. Afterward, a microscope was used for morphological identification. Single selected colonies from the MRS agar plates were transferred into tubes containing MRS broth, followed by incubation under anaerobic conditions at 37 °C for 48 h and stored at −80 °C in MRS broth supplemented with 30% glycerol.

### 5.4. Identification of Bacteria Strains

Five LAB strains dominated in both milk and water kefir grains were cultured in de Mann Rogosa Sharpe (MRS) broth, and then were identified using 16srRNA gene full length sequencing. In detail, a TIANamp bacteria DNA kit (TIANGEN biotech (Beijing) co., ltd) was used to extract the DNA according to standard operating procedures. All bacterial strains amplification was conducted with the following primers: 27F (5′-AGAGTTTGATCMTGGCTCAG-3′) and 1492R (5′-GGTTACCTTGTTACGACTT-3‘). The PCR specific conditions were as follows: lid temperature and volume, 105 °C and 20 μL, respectively. Temperature procedure: 95 °C for 5 min; 94 °C for 30 s; 56 °C for 30 s; 72 °C for 30 s, go to step 2 for 34 cycles, then 72 °C for 10 min and maintained at 4 °C. Afterwards, the products were sent to Sangon Biotech (Shanghai) Co., Ltd. for sequencing. The sequences of the obtained DNA were compared and the percentage of homology was determined by sequences in the NCBI BLAST sequence database (https://blast.ncbi.nlm.nih.gov/Blast.cgi, accessed on: 8 June 2021) [44].

Five LAB strains dominated in both milk and water kefir grains were cultured in de Mann Rogosa Sharpe (MRS) broth, and then were identified using 16srRNA gene full length sequencing. In detail, a TIANamp bacteria DNA kit (TIANGEN biotech (Beijing) co., ltd) was used to extract the DNA according to standard operating procedures. All bacterial strains amplification was conducted with the following primers: 27F (5′-AGAGTTTGATCMTGGCTCAG-3′) and 1492R (5′-GGTTACCTTGTTACGACTT-3‘). The PCR specific conditions were as follows: lid temperature and volume, 105 °C and 20 μL, respectively. Temperature procedure: 95 °C for 5 min; 94 °C for 30 s; 56 °C for 30 s; 72 °C for 30 s, go to step 2 for 34 cycles, then 72 °C for 10 min and maintained at 4 °C. Afterwards, the products were sent to Sangon Biotech (Shanghai) Co., Ltd. for sequencing. The sequences of the obtained DNA were compared and the percentage of homology was determined by sequences in the NCBI BLAST sequence database (https://blast.ncbi.nlm.nih.gov/Blast.cgi, accessed on: 8 June 2021) [44].

### 5.5. Juice Preparation

Three different Brix apple juices were used. One kind of commercial juice having 10.3 °Brix, pH = 3.8 named AJC, made by Beijing Huiyuan Food and Beverage Co., Ltd., was bought in Yangling, and the others were made in a lab, having 12.5 °Brix, pH = 3.8 and 15 °Brix, pH = 4.6, named AJD and AJE, respectively. Apple juice was prepared according to Randazzo, Corona [68] with slight modifications. Apple fruit was bought from a local market in Yangling. Apples were washed with clean water, peeled, cut into small pieces, blended (K600 blender), squeezed, sterilized to 121 °C for 20 min (TSF73002-2018 autoclave) instead of pasteurization, and cooled down to room temperature before being used.

### 5.6. Heat-Inactivated Bacterial Cell Preparation

The LAB that has been identified was transferred to MRS agar and incubated at 37 °C for 26 h; this was repeated twice in the same conditions for better activation purposes. After incubation, bacterial cells were autoclaved at 121 °C for 1 h (TSF733002-2018), centrifuged for 20 min at 4000 rpm at 4° C (ZONKIA; HC-3018-R), and washed three times with distilled water, followed by vacuum freeze-drying at −54 °C for 30 h (MCFD5505; SIM International Group Co. Ltd.) [55].

### 5.7. Patulin Binding Assay

PAT standard stock solution (100 μg/mL) was previously prepared and stored at −20 °C in the NWAFU laboratory. In this study, PAT standard stock solution (100 μg/mL) was diluted to 100 and 200 μg/L. PAT-contaminated apple juice was obtained by adding 100 and 200 μg/L to the juices. The organic solvent ethyl acetate was evaporated from PAT using a water bath (DFS; KW-1000DC) at 40 °C to dryness, then PAT was dissolved in the PAT-free sterile apple juice (10.2, pH = 3.8; 12.5, pH = 3.8; and 15, pH = 4.6) named AJC, AJD, and AJE, respectively [35].

### 5.8. Patulin Releasing Assay

Inactivated bacterial cell (0.1 g) of each bacterium was added to each 10 mL of PAT-contaminated apple juice at different concentrations. Thus, in this study, 0.4 g of inactivated lactic acid bacteria for each strain was added into 40 mL of juice contaminated with PAT. The Erlenmeyer contained PAT-contaminated apple juice, and bacteria cells were placed in a shaker incubator (NRY-2102C) at 120 rev/min at 37 °C for 24 h [35]. Subsequently, the incubation was followed by centrifugation (4000 rpm for 20 min at 30 °C: ZONKIA; HC-3018R). The supernatants were collected for HPLC (high-performance liquid chromatography) analysis, and residual toxin concentrations were dried and characterized FT-IR (Fourier transform infrared spectroscopy).

### 5.9. Extraction and Cleanup

The method of Sajid et al. [35] was used for treated juices. Ethyl acetate was used to extract all assays for three times, both control and juice samples (PAT-contaminated apple juice without bacteria and PAT-contaminated apple juice treated by inactivated lactic acid bacteria), respectively, and then cleaned up by extraction with 4 mL of a 1.5% (*w/v*) sodium carbonate solution. After 2 mL of ethyl acetate and 1 mL of apple juice sample were transferred into a 10 mL tube, it was shaken for 1 min (Model Voltex2 SO25IKA). The cleaned organic phase was then passed over a cotton layer with 2 g of anhydrous sodium sulfate Na2SO_4_ in a 10 mL syringe container and evaporated to dryness (MD200 sample concentrator). The residue was cleaned with 1 mL of deionized water adjusted with acetic acid to pH = 4.0. All samples and the patulin standard solution (Appendix A) were prepared in triplicate and filtered through a pore-size membrane of 0.22μm [69,70].

### 5.10. HPLC Analysis

HPLC analysis was immediately performed. Samples were analyzed by the HPLC system (Shimadzu corporation; L20234608201CD) with a UV absorbance detector and an Altima reversed-phase column C18 (250 mm × 4.6 mm internal diameter, 5 mm particle size) was applied in patulin residue determination. Both wavelength and time detection were set at 276 nm for 20 min, respectively. A 20 μL sample or standard solution of PAT was poured, and HPLC grade water/acetonitrile (90:10, *v/v*) was used as the isocratic mobile phase with a flow rate of 1 mL/min at 40 °C [35].

The percentage of patulin adsorbed by the lactic acid bacteria was calculated by the equation below:(1)$$\%\;\mathrm{Removal}=100\;\times \;[1-(\frac{\mathrm{Peak}\;\mathrm{area}\;\mathrm{of}\;\mathrm{the}\;\mathrm{sample}}{\mathrm{Peak}\;\mathrm{area}\;\mathrm{of}\;\mathrm{the}\;\mathrm{control}})]$$

### 5.11. Characterization of Bacterial Cell Wall

The physical characteristics of inactivated bacteria before and after patulin adsorption were determined by using a transmission electron microscope TEM (TECNAI G2 SPRIT BIO; 03,040,112 FEI COMPANY, Hillsboro, OR, USA); for example, the shapes, cell surface area, and cell volumes were observed and calculated. The morphologies of bacteria powder before and after patulin adsorption were distinguished by a transmission electron microscope TEM (TECNAI G2 SPRIT BIO; 03,040,112 USA FEI COMPANY, Hillsboro, OR, USA), Scanning Electron Microscope SEM Nova Nano-SEM 450 FEI COMPANY (Hillsboro, OR, USA). The cell surface area and cell volume were also calculated. The important functional groups and achievable adsorption sites associated with patulin adsorption were identified by FTIR analysis (Bruker VOTEX70, Karlsruhe Germany), where 2 mg of inactivated kefir grain cells were mixed with 200 mg of KBr (Spectral) grounded in an agate mortar, and then 50 mg of the mixture was pressed into a transparent disc. All IR spectra were recorded at room temperature, and the IR spectra range was 4000 cm^−1^ to 400 cm^−1^. The FTIR experiments were conducted in triplicate.

The calculations of the cell wall volume (V) and surface area (S) of individual cells are given as follows:(2)$$V=\pi h\left[r^{2}-{\left(r-d\right)}^{2}\right]$$
(3)$$S=2\;\pi r^{2}+2\pi rh$$
where *r* is half of the diameter of the cell, *h* is the height of the cell, *p* is Pi, and *d* is the thickness of the cell wall.

### 5.12. Juice Quality Specification

The effects of LAB on juice attributes (sugar content, pH, and aroma) were assessed. The equipment we used to determine pH, Brix, and aroma were a pH meter, Brix refractometer (Melter Poledo: 8029123), and the PEN3 portable E-nose (Win Muster Airsense Analytics Inc., Schwerin, Germany), according to [71]. Before doing our experiment, we tested and equilibrated the E-nose machine using the standard protocol. We diluted every juice specimen ratio to 1:5. We then put it into a 30 mL glass. The sample was equilibrated for 5 min after Teflon/silicone septum was put in the screw cap to allow for headspace enrichment before analysis. Then, the measurement phase lasted for 60 s. Cleaning gas was pumped into the sample gas path for 300 s after each experiment. Sensor intensity was defined as G/G0, where G0 and G are the resistance of the sensor in zero gas and sample gas, respectively.

### 5.13. Statistical Analysis

The experiments were performed using one-way ANOVA in IBM SPSS Statistics software 22.0 (IBM Corp., Armonk, NY, USA) using multiple mean comparisons by Duncan. All of our results were performed in triplicate trials. The results were presented as means ± standard deviations; *p*-values at *p* < 0.05 were considered statistically significant. The principal component analysis (PCA) data were obtained through the Minitab 19.

## Figures and Tables

**Figure 1 toxins-13-00434-f001:**
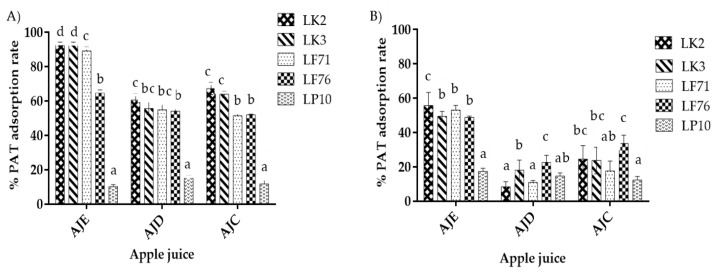
HPLC (high-performance liquid chromatography) outcome after batch adsorption. (**A**,**B**) % of PAT (patulin) adsorption rate from PAT contaminated juice samples at 100 μg/L and 200 μg/L, respectively. AJE, AJD, and AJC are PAT contaminated juice samples after adsorption. LK2, LK3, LF1, LF6, and LP10 represent heat-inactivated lactic acid bacteria strains used to adsorb PAT. Both AJE and AJD are lab-made juices having 15 °Brix, 4.6 pH and 12.5 °Brix, 3.8 pH, respectively. AJC is commercial juice with 10.3 °Brix and 3.8 pH incubated at 30 °C for 26 h. The mean values of triplicate trials are indicated by the bars; error bars show the standard deviation, *p* ≤ 0.05.

**Figure 2 toxins-13-00434-f002:**
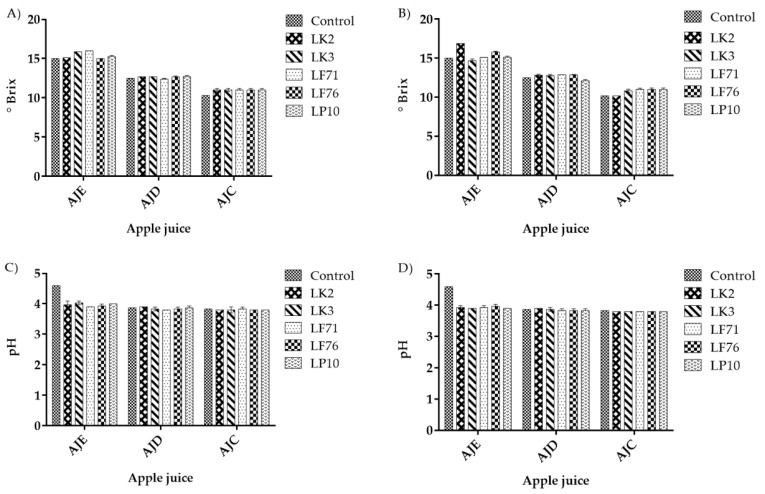
(**A**,**B**) °Brix values of juices contaminated with PAT at 100 and 200 µg/L, respectively, after adsorption; (**C**,**D**) represent the pH value of juice contaminated with PAT at 100 and 200 µg/L after adsorption. The control is the juice without PAT and bacteria. AJE, AD, and AJC are juice samples contaminated with PAT and treated by heat-inactivated lactic acid bacteria strains (after adsorption). LK2, LK3, LF1, LF6, and LP10 represent heat-inactivated lactic acid bacteria strains (0.1 g of bacterial powder) used to adsorb PAT. Both AJE and AJD are lab-made juices having 15 °Brix, 4.6 pH and 12.5° Brix, 3.8 pH, respectively. AJC is commercial juice with 10.3 °Brix and 3.8 pH. Based on one-way ANOVA (analysis of variance) (*p* < 0.05), no significant differences were found between the control and the treated juices.

**Figure 3 toxins-13-00434-f003:**
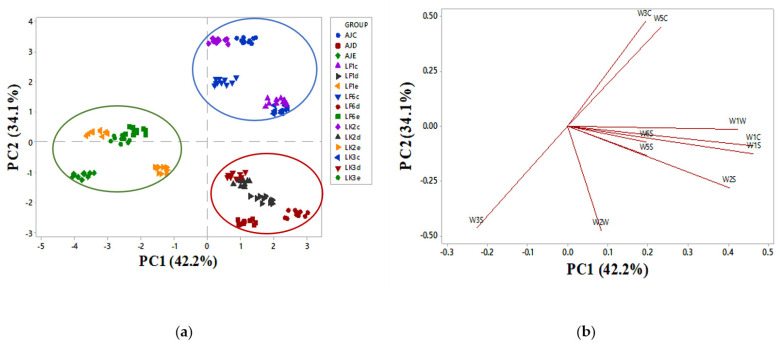
(**a**) The PCA analysis for E-nose. AJC, AJD, and AJE are the controls (juices without PAT and bacteria). LF1, LF6, LK2, and LK3 indicate inactivated lactic acid bacteria strains used to adsorb patulin. The letters c, d, and e indicate that the bacteria have been used in AJC (10.3 °Brix, 3.8 pH, commercial juice), AJD (12.5 °Brix, 3.8 pH, lab-made juice), and AJE (15 °Brix, 4.6 pH, lab-made juice). (**b**) Loading plot for 10 sensors; W1C (sensitive to aromatic compounds), W5S (very sensitive, broad range, react on nitrogen oxides), W3C (sensitive to aromatic compounds of ammonia), W6S (main hydrogen, selectively), W5C (alkanes, aromatic compounds, less polar compounds), W1S (sensitive to methane, broad range), (W1W) reacts on many terpenes, organic sulfides, and inorganic sulfide (H2S), W2S (detects alcohols and partially aromatic compounds, broad range), (W2W) sensitive to aromatic compounds and sulfur organic compounds, and (W3S) sensitive to alkane (methane).

**Figure 4 toxins-13-00434-f004:**
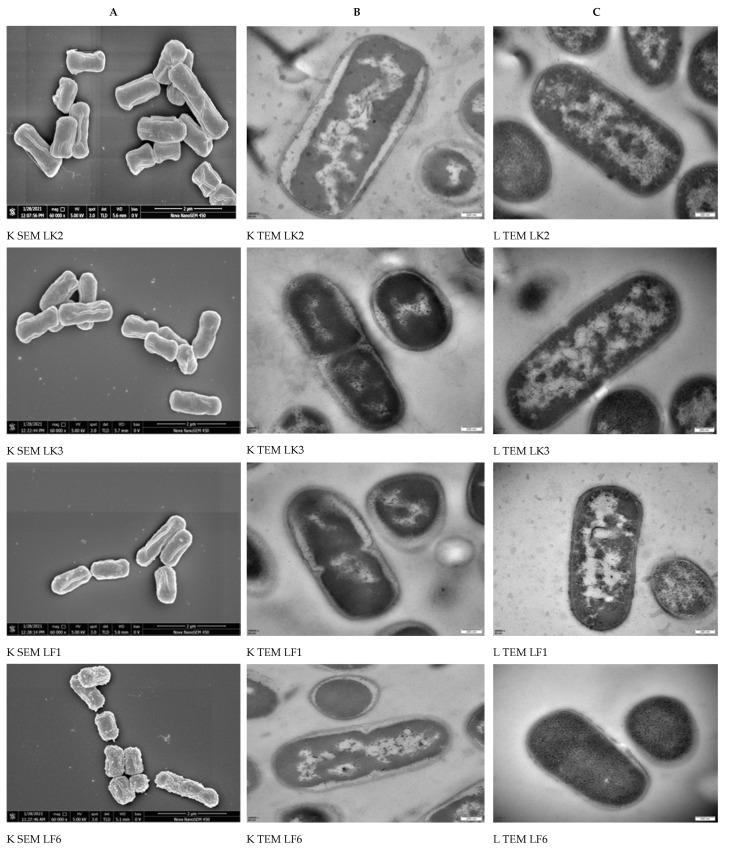
Morphological identification of LAB strains (LK2, LK3, LF1, LF6, and LP10) in AJE at 100 µg/L PAT, where K indicates that heat-inactivated cells adsorbed PAT, and L represents viable cells. (**A**) SEM results, (**B**,**C**) TEM results for heat-inactivated cells adsorbed PAT and viable cells respectively; 60,000× is the magnification of SEMand 48,000× is the magnification of TEM.

**Figure 5 toxins-13-00434-f005:**
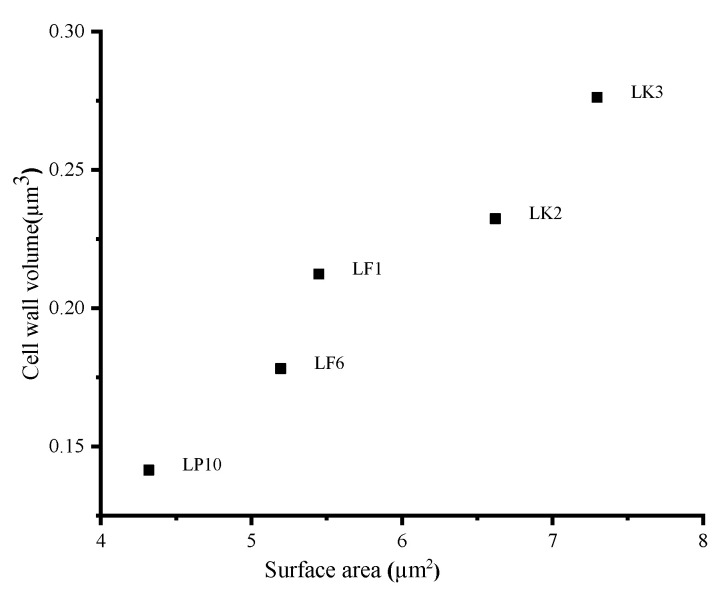
Surface area plotted against the cell wall volume of individual cells in AJE at 100 µg/L.

**Figure 6 toxins-13-00434-f006:**
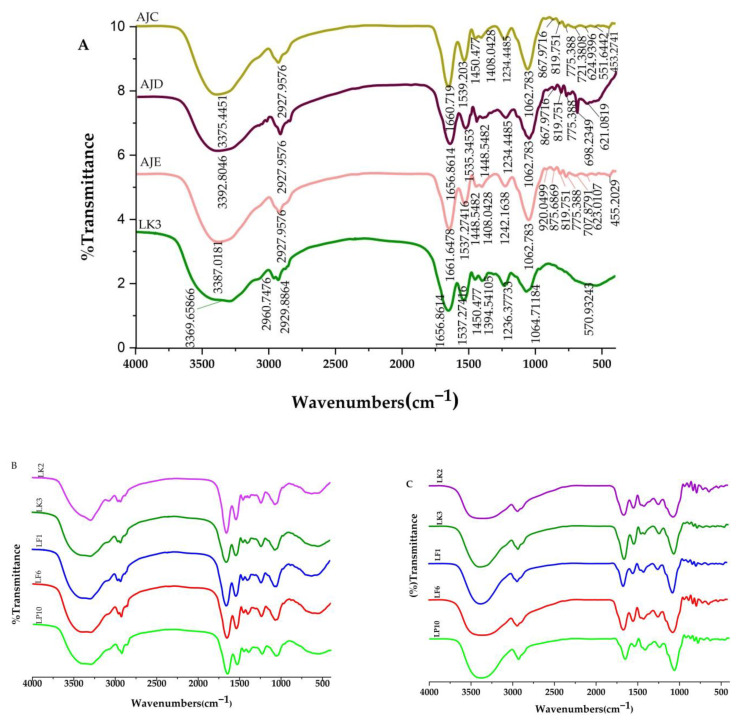
(**A**) FTIR (Fourier transform infrared spectroscopy) spectra of a heat-inactivated strain before and after loading PAT. LK3 represents used lactic acid bacteria strain isolated from Tibetan kefir grains before loading PAT (control). AJE, AJD, and AJC demonstrate LK3 exposed to PAT in apple juices (15 °Brix, pH = 4.6; 12.5 °Brix pH = 3.8; and 10.3 °Brix, pH = 3.8, respectively) after binding at 100 µg/L PAT. (**B**) FTIR spectra of heat-inactivated kefir grain cells LAB (LK2, LK3, LF1, LF6, and LP10) before PAT was loaded and (**C**) FTIR spectra of heat-inactivated kefir grain cells LAB (LK2, LK3, LF1, LF6, and LP10) after loading PAT. Apparently, certain changes in FTIR spectra between the PAT-unexposed and exposed bacterial cells were manifested. Nevertheless, the shape of each peak of the sample was held. Thus, the primary morphology of PAT-exposed bacterial cells was not entirely lost.

**Table 1 toxins-13-00434-t001:** Bacterial characteristics.

Strains	Bacteria Height (nm)	Bacteria Diameter (nm)	Bacteria Thickness (nm)
LK2	1935.45 ± 19.36 ^bc^	885.96 ± 16.73 ^c^	45.39 ± 4.76 ^a^
LK3	2381.66 ± 106.4 ^d^	831.33 ± 94.36 ^bc^	47.65 ± 6.18 ^a^
LF1	2125.54 ± 320.92 ^cd^	701.49 ± 34.13 ^a^	49.45 ± 7.35 ^a^
LF6	1814.23 ± 2.97 ^ab^	754.31 ± 35.37 ^ab^	44.02 ± 1.39 ^a^
LP10	1552.78 ± 138 ^a^	717.44 ± 36.87 ^a^	42.73 ± 1.6 ^a^

Cell wall sizes and thicknesses. Different letters indicate significant differences within the column of height and diameter. There were no significant differences between bacteria thicknesses (*p* < 0.05).

**Table 2 toxins-13-00434-t002:** Fourier transform infrared spectroscopy spectra observed from PAT-unexposed and exposed bacterial cells.

Functional Groups	Wavenumber (cm^−1^)				
	LK2	LK3	LF1	LF6	LP10
O-H/N-H stretching	(3325.29567) 3377.374	(3369.65866) 3387.0181	(3363.8721) 3371.5875	(3367.7298) 3371.5875	(3371.5874) 3383.1604
C-H stretch	(2964.60525) *	(2960.74759) *	(2964.60525) *	(2960.74759) *	(*) *
C-H stretch	(2929.88638) 2935.6729	(2929.88638) 2937.6017	(2929.88638) 2929.8864	(2926.02873) 2935.6729	(2926.02873) 2931.8152
C=O amide 1	(1658.7902) 1653.00371	(1656.8614) 1653.0037	(1658.7902) 1654.9325	(1651.07488) 1656.8614	(1645.2884) 1653.0037
N-H amide 2	(1537.27416) 1539.203	(1537.27416) 1537.2742	(1531.4877) 1539.203	(1539.20298) 1537.2742	(1535.34533) 1541.1318
N-O asymmetric stretch	(*) *	(*) 1448.5482	(*) *	(*) *	(*) *
O-H deformation	(1456.26348) 1411.9005	(1450.477) 1408.0428	(1452.4058) 1409.9717	(1454.33465) 1409.9717	(1454.33465) 1406.114
C-C stretch	(1382.96809) *	(1394.54105) *	(1394.54105) *	(1398.3987) *	(1396.46988) *
C-N amide 3	(1238.30616) 1240.235	(1236.37733) 1242.1638	(1234.44851) 1236.3773	(1236.3773) 1236.3773	(1238.3061) 1234.44
C-O polysaccharides	(1064.71184) 1062.783	(1064.71184) 1062.783	(1060.85419) 1062.783	(1062.78301) 1060.8542	(1060.85419) 1058.9254
O-H bend	(*) 923.9076	(*) 920.0499	(*) 923.9076	(*) 921.9787	(*) 918.1211
O-H bend	(*) 869.9004	(*) 875.6869	(*) 871.8293	(*) 864.114	(*) 862.18513
C-Cl stretch	(*) 815.8933	(*) 819.751	(*) 819.751	(*) 819.751	(*) 821.6798
C-Cl stretch	(*) 773.4591	(*) 775.388	(*) 775.388	(*) 779.2456	(*) 773.459144
C-Br Alkyl Halide	(*) 704.0214	(*) 707.8791	(*) 709.8079	(*) 704.0214	(*) 704.021415
C-Br Alkyl Halide	(*) 630.726	(*) 623.0107	(*) 630.726	(*) 630.726	(*) 628.7972
C-Br Alkyl Halide	(592.149519) *	(570.932435) *	(599.864822) *	(582.50539) *	(580.576564) *
C-Br Alkyl Halide	(*) 516.9253	(*) *	(*) *	(*) 516.9253	(*) 516.9253
C-I Alkyl Halide	(*) 451.3452	(*) 455.2029	(*) 455.2029	(*) 449.4164	(*) 455.2029

The observed wavenumbers summary for the five LAB strains used to detoxify PAT-contaminated apple juice AJE (15 °Brix, pH = 4.6). The enclosed values with parentheses signify the controls (PAT-unexposed, heat-inactivated LAB strains). * Symbolizes the bands not observed. The figure representing the corresponding peaks for the abovementioned wavenumbers was given in Appendix A.

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
