# Peer review of "Adsorption Mechanism of Patulin from Apple Juice by Inactivated Lactic Acid Bacteria Isolated from Kefir Grains"

_toxins, 2021, doi:10.3390/toxins13070434_

Round 1
Reviewer 1 Report
Dear Authors,
This manuscript reports about the identify and characterize an excellent adsorbent to remove patulin from apple juice efficiently and to assess its adsorption mechanism.
Research is interesting as well as has some scientific value.
Major flaw:
- The current studies should be summarized in the Introduction section to provide comprehensive background for the topic that the paper pertains. The authors dont quote recent studies such as:
Xiangfeng Zheng, Wanning Wei, Shengqi Rao, Lu Gao, Huaxiang Li, Zhenquan Yang. (2020). Degradation of patulin in fruit juice by a lactic acid bacteria strain Lactobacillus casei YZU01. Food Control, Vol. 112, 107147, ISSN 0956-7135, https://doi.org/10.1016/j.foodcont.2020.107147.
Although the study deserves some merit, the novelty of the work is very average.
Minor comments
- Lack of information about PCA in the section Satistical analysis.
- Figure 1- lack p – value.
- Line 360 : there was no information about manufacture of commercial juice.
- Not described how the bacterial strains were added to apple juice.
- With such big amount of performed analyses, flow diagram could be helpful.
In my opinion, this manuscript isn’t appropriate for publication in Journal – Toxins.
Author Response
Response to Reviewer 1 Comments
This manuscript reports about identify and characterize an excellent adsorbent to remove patulin from apple juice efficiently and to assess its adsorption mechanism.
Research is interesting as well as has some scientific value.
- Major flaw:
Point 1. The current studies should be summarized in the Introduction section to provide a comprehensive background for the topic that the paper pertains. The authors don't quote recent studies such as:
Xiangfeng Zheng, Wanning Wei, Shengqi Rao, Lu Gao, Huaxiang Li, Zhenquan Yang. (2020). Degradation of patulin in fruit juice by a lactic acid bacteria strain Lactobacillus casei YZU01. Food Control, Vol. 112, 107147, ISSN 0956-7135, https://doi.org/10.1016/j.foodcont.2020.107147.
Response1; The current studies in the introduction section were summarized and the recent studies such as the above-suggested study were cited in the submitted manuscript line 43 to 53 as suggested
- Minor comments
Point 1. Lack of information about PCA in the section statistical analysis.
- Response1: PCA information was added in the section of satistical analysis as you suggested in line 533 and 534
Point 2. Figure 1- lack p-value.
- Response 2: p-value was added in Figure 1 as suggested
Point 3. Line 360: there was no information about the manufacture of commercial juice.
- Response 3: The information about the manufacture of commercial juice was provided in the submitted manuscript line 443 and 444 as you suggested
Point 4. Not described how the bacterial strains were added to apple juice.
- Response 4: The information about how the bacterial strains were added to apple juice was described in the submitted manuscript line 468 and 469 as suggested
Point 5. With such big amount of performed analyses, a flow diagram could be helpful.
- Response 5: The flow diagram was provided as you suggested (Figure 7).
WE EXTREMELY THANK THE REVIEWER FOR THE TIME AND CONSTRUCTIVE COMMENTS
Reviewer 2 Report
The manuscript entitled "Adsorption Mechanism of Patulin from Apple Juice by Inactivated Lactic Acid Bacteria Isolated from Kefir Grains" hase been revised and found it is interesting but needs some improvment to be suitable for publication as follow:
-
- Methodolgy
- The experental design is missed, the author mus provide clear and understandable experemintal design.
- Author always mentioned "slight modifications", which mofifications have been made by authors in the mthods?
- What are the difference between three types of juices (AJC, AJD & AJE)?
- in ptulin pinding assay- author used 3types of patulin and tow different concentrations (100 and 200 µg/L)?
- What are the differences between (Contaminated apple juice and patulin contaminated apple juice)?
- Author must expalin the positive and nigative conterol ( they are unclear in the text)
- In line 394- " After 2 mL ethyl acetate and 1 mL, apple juice solution was transferred into a 10 mL tube, it was shaken for 1 min (Model Voltex2 SO25IKA)." 1 ml of what??
- Results and disscusion
- In section 2.1. Strain identification- where are the results of molecular identification?- the results must be presented.
- In (fig. 1), where are the positive and nigative control? wher the results of the second concnertation (200μg/l)
- In Table 1. where are the positive and nigative control? the results for which concnertation (100 or 200μg/l)?
- Fig. 3 and 4 for which concentration 100 or 200200μg/l) and which type of juices?
- Fig 5. where the positive and nigative control? the praphs of FTIR for different spices of pacteria but it doesn't clear for what type of juices? in B&C must diferenciated which one before and which one after?
- Methodolgy
Author Response
Response to Reviewer 2 comments:
The manuscript entitled "Adsorption Mechanism of Patulin from Apple Juice by Inactivated Lactic Acid Bacteria Isolated from Kefir Grains" has been revised and found it is interesting but needs some improvement to be suitable for publication as follow:
- Methodology
Point 1. The experimental design is missed, the author must provide a clear and understandable experimental design.
- Response 1. A clear and understandable experimental design was provided as suggested ( point 1). Find experimental design in the supplementary materials.
Point 2. The authors always mentioned "slight modifications", which modifications have been made by authors in the methods?
- Response 2. The slight modifications made in sections 5.3 and 5.5 were mentioned in the submitted manuscript as suggested. Those slight modifications are: In section 5.3 we used a mortar and pestle instead of using a stomacher line 413. While in section 5.5 we used applied sterilization instead of pasteurization line 449.
Point 3. What is the difference between the three types of juices (AJC, AJD & AJE)?
- Response 3. The differences between AJC, AJD & AJE juices were clearly explained as suggested (point 3). In line 443 to 445
- AJC is commercial apple juice made by Beijing Huiyuan Food and Beverage Co., Ltd, having 10.3 °Brix and 3.8 pH.
- AJD & AJE are both lab-made apple juices having 12.5 °Brix, 3.8pH and 15 °Brix, 4.6 pH respectively.
Point 4. in patulin binding assay- author used 3types of patulin and two different concentrations (100 and 200 µg/L)?
- Response 4. Patulin binding assay in the section regarding the types of patulin concentration used was clearly explained as suggested in line 459 to 461
- Actually, In our study, we used to100 and 200μg/L PAT concentration in apple juice.
- 100μg/mL is the standard stock solution which was diluted to100 and 200μg/L before use.
Point 5. What are the differences between (Contaminated apple juice and patulin-contaminated apple juice)?
- Response 5.There are no differences between contaminated apple juice and patulin-contaminated apple juice it could be a typing error. A well-used terminology is PAT contaminated apple juice, the word has been corrected in the line 478and 479
Point 6. The author must explain the positive and negative control ( they are unclear in the text)
- Response 6. The terms “positive and negative control have been changed to better understanding the study, We used the terms control and samples instead. Line 478
Point 7. In line 394- " After 2 mL ethyl acetate and 1 mL, apple juice solution was transferred into a 10 mL tube, it was shaken for 1 min (Model Voltex2 SO25IKA)." 1 ml of what??
- Response 7. This 1ml meant 1ml of juice sample, which has been corrected in the submitted manuscript. Line 481
- Results and discussion
Point 1. In section 2.1. Strain identification- where are the results of molecular identification?- the results must be presented.
- Response 1. The results of molecular identification in section 2.1. was presented as suggested. Line 429 to 431
Point 2. In (fig. 1), where are the positive and negative control? where the results of the second concentration (200μg/l)
- Response 2. The results of the second concentration (200μg/l) were provided as suggested. Figure 1.
- To better representing the results, the previous figure has changed to the new one indicating both results.
- % Removal =100 × [1- ()]
- According to the above-used formula, no control value can be calculated, Figure 1. Indicate the % adsorbed, thus no adsorption was done in control because control represents apple juice with PAT but without bacteria
Point 3. In Table 1. where is the positive and negative control? the results for which concentration (100 or 200μg/l)?
- Response 3. To ensure a good presentation of the results (100 or 200μg/l), table 1 was changed to figure 2. In this section, controls were well explained. The actual figure represents the results of 100 or 200μg/l PAT concentration in all used juices and bacteria.
Point 4. Fig. 3 and 4 for which concentration 100 or 200200μg/l) and which type of juices?
- Response 4. In section 2.5 about Morphological Analysis of Bacterial Cells by SEM and TEM, The concentration and type of juice related to the figures were provided as suggested. (point 4) AJE at 100 µg/L. Line 237
Point 5. Fig 5. where the positive and negative control? the graphs of FTIR for different spices of bacteria but it doesn't clear for what type of juices? in B&C must be differentiated which one before and which one after?
- Response 5. A question asked by the reviewer regarding the type of juice used in the FTIR section was well answered (point5) 265 and 266
- (B) FTIR spectra of heat-inactivated kefir grain cells LAB (LK2, LK3, LF1, LF6 and LP10) before PAT was loaded and (C) FTIR spectra of heat-inactivated kefir grain cells LAB (LK2, LK3, LF1, LF6 and LP10) after loading patulin PAT at 100µg/L, line 267
WE EXTREMELY THANK THE REVIEWER FOR THE TIME AND CONSTRUCTIVE COMMENTS
Reviewer 3 Report
General remarks:
Patulin can be abbreviated as PAT. It appears in abbreviated form only in the conclusion section.
Bacterial names must be written in capital letters (Lactobacillus instead of lactobacillus)
In general, results are not clearly presented. Tables and figures must be provided in the corresponding section and information must be understood without consulting the methodology section.
Concise remarks:
Abstract
Line 13: change the symbol of degree into °
Introduction
Line 24: Penicillium instead of penicillium
Lines 28-29: In my opinion, this information could be detailed highlighting recent information on some genotoxic mechanisms exerted by patulin. Moreover, this information can be included in another paragraph as is not related neither with previous nor with the following paragraph.
Lines 35-45: The authors said that “daily intake limits have been established”; however, the information included in this paragraph is related to maximum levels (MLs) established for certain foodstuffs, but no information on the Tolerable Daily Intake (TDI) is given. In addition, according to the EFSA Scientific Opinion on Patulin (2020), the established TDI is no longer valid and a Margin of Exposure (MoE) approach must be applied, thus setting two BMDL for patulin. In my opinion the authors must include this information if they are talking about the daily intake or they can omit this information if they are just talking about MLs.
Lines 50-51: This sentence is not related to information given in this paragraph and is not developed in next paragraph.
Lines 66-68: However, there are not enough citations showing the adsorption mechanism with samples having different °Brix were provided and the adsorption mechanism of patulin needs to be more clear [29].
Line 71: °Brix
Results
Figure 1: It must be indicated the pH for the AJC.
Section 2.2 deals on patulin absorption, in my opinion the title of the section must be changed into another more appropriate.
Line 99: 15 °Brix
Lines 112, 113, 123 and 135: °Brix This is a common mistake along the manuscript, please revise all the manuscript and try to be consistent.
Section 2.3 only includes one table and one figure, both of them cited in another section. Tables and figures must be included in corresponding sections.
Line 136: Figure 2A and B must be in brackets.
Line 139: E-nose has been performed to determine…
Discussion
Line 236: and clarity. ([22, 29].
References:
References and cites are not consistent. When cites are included in the text, it appears the surname of the two first authors and in the reference section it appears the surname of the first author and “et al.”. I suggest revising the journal guidelines in order to be consistent.
Author Response
Response to Reviewer 3 Comments:
- General remarks:
Point 1. Patulin can be abbreviated as PAT. It appears in abbreviated form only in the conclusion section.
- Patulin was abbreviated to PAT ( point 1), Bacterial names were written in capital letters( point 2),
Point 2. Bacterial names must be written in capital letters (Lactobacillus instead of lactobacillus)
- Response 2. Bacterial names were written in capital letters( point 2),
In general, results are not clearly presented. Tables and figures must be provided in the corresponding section and information must be understood without consulting the methodology section.
- Response 3. The results were clearly presented, tables and figures are provided in the corresponding section (point 3)
- Concise remarks:
Abstract
Point 1. Line 13: change the symbol of degree into °
- Response 1. Changing of the symbol of degree into ° was made in the whole text
- Introduction
Point 1. Line 24: Penicillium instead of penicillium
- Response 1. Bacterial names were written in capital letters, all bacteria names were written in the capital in the whole text
Point 2. Lines 28-29: In my opinion, this information could be detailed highlighting recent information on some genotoxic mechanisms exerted by patulin. Moreover, this information can be included in another paragraph as is related neither with the previous nor with the following paragraph.
- Response 2. Genotoxic mechanisms exerted by patulin have been detailed according to recent information. Line 29 to 38
Point 3. Lines 35-45: The authors said that “daily intake limits have been established”; however, the information included in this paragraph is related to maximum levels (MLs) established for certain foodstuffs, but no information on the Tolerable Daily Intake (TDI) is given. In addition, according to the EFSA Scientific Opinion on Patulin (2020), the established TDI is no longer valid and a Margin of Exposure (MoE) approach must be applied, thus setting two BMDL for patulin. In my opinion, the authors must include this information if they are talking about the daily intake or they can omit this information if they are just talking about MLs.
- Response 3. The word daily intake was omitted from the text as suggested because the information was related to maximum levels, not the daily intake limit
Point 4. Lines 50-51: This sentence is not related to information given in this paragraph and is not developed in next paragraph.
- Response 4 Lines 50-51 the sentence was omitted from the manuscript as suggested
Point 5. Lines 66-68: However, there are not enough citations showing the adsorption mechanism with samples having different °Brix were provided and the adsorption mechanism of patulin needs to be more clear [29].
- Response 5 Lines 66-68: In this sentence; However, there are not enough citations showing the adsorption mechanism with samples having different °Brix were provided and the adsorption mechanism of patulin needs to be more clear. The word “were provided” was removed from the text as suggested ..
Line 71: °Brix
- Results
Point 1. Figure 1: It must be indicated the pH for the AJC.
- The pH was indicated for the AJC in figure 1.
Point 2. Section 2.2 deals with patulin absorption, in my opinion, the title of the section must be changed into another more appropriate one.
- Response 2. In section 2.2, the title “Detection and Quantification of Patulin in Treated Juice” was changed to HPLC Analysis. Line 157
Point 3. Line 99: 15 °Brix
- Response 2 Line 99, 112, 113, 123, and 135: °Brix has been reviewed in the whole text as suggested (points 3 and 4).
Point 5. Section 2.3 only includes one table and one figure, both of them cited in another section. Tables and figures must be included in corresponding sections.
- Response 5 In section 2.3, the table and figure were cited in the corresponding sections (point 5).
Point 6. Line 136: Figure 2A and B must be in brackets.
- Response 6 Figure 2A and B was put in brackets( point 6). Line 196
- Figure has changed the position to 3
Point 7. Line 139: E-nose has been performed to determine…
- Response 7 E-nose has been performed to determine ……. ( point 7)The sentence was completed as the following; E-nose has been performed to determine apple juice aroma. Line 199
- Discussion
Point 1. Line 236: and clarity. ([22, 29].
- Response 1. The reference was corrected (point 1).line 310
- References:
References and cites are not consistent. When cites are included in the text, it appears the surname of the two first authors and in the reference section it appears the surname of the first author and “et al.”. I suggest revising the journal guidelines in order to be consistent.
- Response The references were cited according to the journal guidelines
WE EXTREMELY THANK THE REVIEWER FOR THE TIME AND CONSTRUCTIVE COMMENTS
Round 2
Reviewer 1 Report
Dear Authors,
This manuscript reports about the identifying and characterizing an excellent adsorbent to remove patulin from apple juice efficiently and to assess its adsorption mechanism.
The majority of reviewer’s comments were taken into account by Authors. After reading the new version of manuscript, I have no objections. In my opinion, this text is appropriate for publication in Journal – Toxins.
Reviewer 2 Report
The manuscript entitled "Adsorption mechanism of patulin from apple juice by inactivated lactic acid bacteria isolated from kefir grains" hase been rerevised and found that the aouthors considered all comments and manuscript has been improved the . So, i think the manuscript in the present fromat is suitable for publications.
Reviewer 3 Report
The manuscript have been improved by adding some missing studies and all the comments suggested have been included.